# Thermal Effects on the Population Parameters and Growth of *Acyrthosiphon pisum* (Harris) (Hemiptera: Aphididae)

**DOI:** 10.3390/insects11080481

**Published:** 2020-07-29

**Authors:** Jeong Joon Ahn, Jum Rae Cho, Jeong-Hwan Kim, Bo Yoon Seo

**Affiliations:** 1Research Institute of Climate Change and Agriculture, National Institute of Horticultural and Herbal Science, RDA, 281, Ayeon-ro, Jeju 63240, Korea; j2ahn33@korea.kr; 2Crop Protection Division, Department of Agro-food Safety and Crop Protection, National Institute of Agricultural Sciences, RDA, 166, Nongsaemgmyeong-ro, Wanju-gun, Jeollabuk-do 55365, Korea; jrcho82@korea.kr (J.R.C.); kim9@korea.kr (J.-H.K.)

**Keywords:** *Acyrthosiphon pisum*, life table, population projection, temperature

## Abstract

The pea aphid *Acyrthosiphon pisum* (Harris) (Hemiptera: Aphididae) is a cosmopolitan and polyphagous species. An evaluation of *A. pisum*’s demographic parameters and growth was carried out after rearing aphids on faba bean plants (*Vicia faba*) under five different temperature conditions (10 °C, 15 °C, 20 °C, 25 °C and 30 °C). We analyzed the raw life history data, including developmental time, survival, longevity and reproduction, using an age-stage, two-sex life table to consider variable developmental rates among individuals. The population fluctuation of *A. pisum* determined the stage-specific population structure and potential population growth under different temperature conditions. *A. pisum* individuals developed successfully from nymphs to adults at all temperatures in this study. The developmental rate of *A. pisum* increased as the temperature increased. Our results indicated that *A. pisum* showed a higher pre-adult mortality, lower total fecundity and a negative intrinsic rate of increase at 30 °C. The highest intrinsic rate of increase (0.30) and finite rate of increase (1.35) were observed at 25 °C. Comparisons of population parameters and their analytical methods between different *A. pisum* populations from other geographic areas are also discussed.

## 1. Introduction

The pea aphid *Acyrthosiphon pisum* Harris (Hemiptera: Aphididae) is a cosmopolitan and polyphagous species whose life cycle varies depending on geographic location [1]. Pea aphids suck sap from young plant shoots, which directly causes the plants to wilt and dry up, and indirectly causes sooty mold to grow on plant leaves, which reduces photosynthesis due to honeydew excretion [2,3]. Pea aphids can also transmit plant viruses, such as pea enation mosaic virus (PEMV), bean leafroll virus, bean yellow mosaic virus and soybean dwarf virus (SbDV), which can impede the cultivation of crops like *Vicia faba* and *Pisum sativum* and cause economic losses [4,5,6,7].

Climate change has attracted worldwide attention, and it has become important to understand the incidence and severity of climate ‘presses’ and ‘pulses’, such as extreme fluctuations of temperature, rainfall, droughts and floods [8,9], and its complex effects, such as range changes, life-history traits, population dynamics and trophic interactions, on insect pests in agriculture [10]. Aphids are one of the most important pests of numerous agricultural crops worldwide, and have a high reproductive capacity coupled with rapid life stage conversion. Their rates of development, survival, longevity and reproduction depend on biotic (e.g., host plants) and abiotic factors (e.g., temperature, humidity and photoperiod). Temperature is a notable abiotic factor that facilitates the development and fecundity of the insects [11,12,13,14]. Many studies have introduced temperature-dependent development models based on the developmental rates of insects at different temperatures [15,16,17,18,19,20,21,22]. The relationship between temperature and the developmental rate has been explained by biophysical processes and mathematical methods. Taking variable developmental rates among individuals into consideration, Chi and Liu [23] and Chi [24] developed the age-stage, two-sex life table theory, which incorporates the stage differentiations and both sexes. Life table studies are more inclusive, and provide more information on population parameters and the growth of insects based on biotic or abiotic factors than temperature-dependent development models [25,26,27].

Life table analysis is crucial for analyzing and understanding the overall biological performance (e.g., development, survival, reproduction and population growth) of an insect pest under different environmental conditions. Thus, in this study, we collected raw life history data of *A. pisum* at five different constant temperatures, ranging from 10 °C to 30 °C (5 °C interval), and analyzed population parameters and growth using the age-stage, two-sex life table theory to determine the effect of temperature on the development, survival and reproduction of *A. pisum*. Although the *A. pisum* populations evaluated in this study reproduce by thelytokous parthenogenesis, we used the age-stage, two-sex life table theory to precisely show the stage differentiation among individuals, and then used the life tables to project the population growth at different temperatures. A few studies have conducted experiments examining the effects of temperature on the development and reproduction of *A. pisum* at different temperatures, derived general relationships between temperature and its development, and constructed life-tables for the species when reared on *P. sativum* [28] and *V. faba* [29], respectively. Although previous studies have illustrated age-specific survivorship, fecundity and population parameters using the age-stage, two-sex life table methods, they did not address the population growth projection and stage growth rate, or the contribution of stage differentiation to the survivorship and reproduction of *A. pisum*. Therefore, the main objectives of this study were to (1) determine the effects of temperature on the biology and development of *A. pisum*, (2) describe the life table parameters for the aphid, and (3) project the population growth and stage-specific growth rates for the species.

## 2. Materials and Methods

### 2.1. Insect Colony

Apterous *A. pisum* adults and nymphs were collected in 2009 from a pea cultivation area in Nonsan-si, Republic of Korea (36°12′ N, 127°09′ E), and kept in an insectary of the National Institute of Agricultural Sciences, Wanju-gun, Republic of Korea (35°49′ N, 127°02′ E). Apterous adults and nymphs were transferred onto faba bean (*V. faba*) leaves in an acrylic plastic container (55 cm × 50 cm × 30 cm (width × length × height)). The faba bean plants were grown in a plastic pot (17 cm × 12 cm (diameter × height)) until they reached 10 cm in height for experiment in the greenhouse. Colonies of *A. pisum* were maintained for about five years in the insectary at 25 ± 2 °C and 60 ± 10% relative humidity under a 16:8 h (light:dark) cycle.

### 2.2. Life Table Study

Newly produced nymphs (<12 h) were randomly selected with a brush and individually placed on a faba bean leaf in a cylindrical plastic cage (5.5 cm × 2.0 cm (diameter × height)) with a lid, which had an air hole measuring 1.3 cm (diameter) that was covered with a 0.05-mm hole mesh. A 1.7% agar in water solution filled the bottom of the cage (0.8 cm in height), and a faba bean leaf was placed on the agar gel to protect the leaves from drying out. Fresh faba bean leaves were supplied as a food source for the nymphs and were replaced with fresh ones as needed. 

The life table study was conducted at temperatures of 10.0 °C, 15.0 °C, 20.0 °C, 25.0 °C and 30.0 °C. Each cage was randomly assigned one of these temperatures, which were adjusted to remain constant within the cylindrical plastic cages in a multi-chamber incubator (NK System, Bio Multi Incubator JP/LH-30-8CT, Nippon Medical & Chemical Instrument Co. LTD., Osaka, Japan). The nymphs were allowed to develop to adults within the cages (each with a constant temperature), and the developmental durations and survivorship for each nymphal stage were recorded. The developmental stage of each aphid nymph was observed visually at 24-h intervals until the nymph reached the adult stage. The presence of aphid exuviae was used as evidence of molting and the transition to the next developmental stage. After the emergence of adults, the survival, longevity and reproductive performances of the individuals were observed daily until the death of each individual. Sample sizes were more than 50 nymphs per temperature treatment.

### 2.3. Life Table Data Analysis

The *A. pisum* life history data were analyzed based on the age-stage, two-sex life table theory [23] and the method described by Chi [24] using the computer program TWOSEX-MSChart [30]. The age-specific survival rate (*l_x_*) is defined in Equation (1):(1)$$l_{x}=\sum_{j=1}^{k}S_{xj}$$
where *k* is the number of stages. The survival rate (*s_xj_*) is defined as the probability that a newly born individual will survive to age *x* and stage *j*. The *s_xj_* curves show the initial age *x* of the survival curve of stage *j*, the final age *x* of the survival curve of stage *j*, the mortality of stage *j* and the age of the individuals that died in stage *j*. Thus, the *s_xj_* curves can show death, and the beginning and the end of a stage. 

The age-specific fecundity (*m_x_*) is calculated to consider individuals of different stages at age *x* and is defined in Equation (2):(2)$$m_{x}=\frac{\sum_{j=1}^{k}S_{xj}f_{xj}}{\sum_{j=1}^{k}S_{xj}}$$

The age-stage-specific fecundity (*f_xj_*) is the number of nymphs produced by an adult female of age *x*. The net reproductive rate (*R_o_*) is defined as the total number of offspring that an individual can produce during its lifetime, and is shown in Equation (3):(3)$$R_{o}=\sum_{x=0}^{\infty }l_{x}m_{x}$$

The intrinsic rate of increase (*r*) is calculated using the Euler–Lotka formula with age indexed from day “0” [31] and is shown in Equation (4):(4)$$\sum_{x=0}^{\infty }e^{-r\left(x+1\right)}l_{x}m_{x}=1$$

The finite rate of increase (λ) is calculated with Equation (5):(5)$$\lambda =e^{r}$$

The mean generation time (*T*) defines the period for which a population needs to undergo an *R_o_*-fold increase in its original size as the population growth rate settles down to the intrinsic rate of increase, and is shown in Equation (6):(6)$$T=\frac{\ln R_{o}}{r}$$

The age-stage-specific life expectancy (*e_xj_*) is the expected life span of an individual of age *x* and stage *j*, and is calculated with Equation (7): (7)$$e_{xj}=\sum_{i=x}^{\infty }\sum_{y=j}^{k}{s^{\prime }}_{iy}$$
where $e_{xj}$ is calculated by assuming ${s^{\prime }}_{iy}$ = 1, and ${s^{\prime }}_{iy}$ is defined as the probability that an individual of age *x* and stage *j* will survive to age *i* and stage *y*. The age-stage reproductive value (*v_xj_*) describes the contribution of an individual of age *x* and stage *j* to the future population [32], and is shown in Equation (8):(8)$$v_{xj}=\frac{e^{r\left(x+1\right)}}{s_{xj}}\;\sum_{i=x}^{\infty }e^{-r\left(i+1\right)}\sum_{y=j}^{k}{s^{\prime }}_{iy}f_{iy}$$

The adult pre-oviposition period (APOP) is determined by the pre-oviposition period based on the female adult age. The total pre-oviposition period (TPOP) is defined as the total amount of time from birth to the initial oviposition. The standard errors of the developmental duration, longevity, fecundity and population parameters were calculated using a bootstrap method with 100,000 replicates. The statistical differences among treatments were analyzed with a paired bootstrap test at the 5% significance level [33]. The bootstrap method was contained in the computer program TWOSEX-MSChart. 

### 2.4. Population Projection

The TIMING-MSChart program was used to predict and compare the population growth and age-stage structure of *A. pisum* in the different temperature treatments [34]. A total of 10 newborn nymphs was used as the starting density to simulate the population growth of each treatment. The files obtained from TWOSEX-MSChart were used to operate the TIMING-MSChart. A common logarithm was used to describe the population growth of stage *j* from time *t* to *t* + 1. The growth rate of stage *j* from time *t* to *t* + 1 was calculated using a natural logarithm [26]:(9)$$\varphi_{j,\;t}=\ln\left(\frac{n_{j,\;t+1}+1}{n_{j,\;t}+1}\right)=\ln\left(n_{j,\;t+1}+1\right)-\ln\left(n_{j,\;t}+1\right)$$

In the calculation process, *n_j,t_* + 1 and *n_j,t+_*_1_ + 1 were used because the transformation of the logarithm is impossible when the number of individuals at a certain stage becomes 0 (*n_j,t_* = 0 or *n_j,t_*_+1_ = 0).

## 3. Results

*Acyrthosiphon pisum* showed four nymphal instars, and was able to successfully develop from the first instar to the adult stage at all temperatures tested in this study. The parameters related to the development of each life stage and the reproduction of the adult females at the five different constant temperatures are summarized in Table 1. The pre-adult duration ranged from 6.5 d at 30 °C to 21.3 d at 10 °C. The longevity and oviposition periods of the adult females decreased with increasing temperatures. There was a significant difference among the TPOPs under the different temperature conditions, and the TPOP decreased from 26.4 d at 10 °C to 8.1 d at 25 °C, and 10.5 d at 30 °C. The shortest reproductive period (2.0 d) and the lowest fecundity (4.5 nymphs/female) were observed at 30 °C. The highest mean fecundity was recorded at 15 °C (74.9 nymphs/female), followed by 20 °C (62.5 nymphs/female). The total longevity of *A. pisum* decreased from 59.1 d to 14.8 d at 10 °C and 30 °C, respectively. The probability that a newly emerged nymph would survive to the adult stage was 0.78, 0.95, 0.82, 0.66 and 0.15 at 10 °C, 15 °C, 20 °C, 25 °C and 30 °C, respectively, and those values coincide with the respective proportions of reproductive adult female individuals (*N_f_*/*N*) in the cohort (Table 2).

The age-stage specific survival rate (*s_xj_*) is the probability that a newborn aphid will survive to age *x* and stage *j* (Figure 1). The curves of each life stage show the survivorship, stage differentiation and overlap among stages due to the variable developmental rates among individuals. Adults emerged at 14 d, 10 d, 7 d, 5 d and 6 d, and survived until 91 d, 63 d, 47 d, 21 d and 16 d at 10 °C, 15 °C, 20 °C, 25 °C and 30 °C, respectively.

The age-specific survival rate and fecundity of *A. pisum* are presented in Figure 2 and Figure 3, respectively. The age-specific survival rate (*l_x_*) is the sum of *s_xj_* at each age *x*, and is thus the simplified version of *s_xj_* in Figure 1. The curves sharply dropped for the earlier ages at 20 °C to 30 °C, compared to those of the other temperature treatments. The fecundity curve, *m_x_*, ended at age 88 d, 45 d, 34 d, 20 d and 13 d at 10 °C, 15 °C, 20 °C, 25 °C and 30 °C, respectively. The durations from the first to the last reproduction at five different temperatures ranged from 5 d (30 °C) to 70 d (10 °C). The post-reproductive period was the shortest at 25 °C, with 2 d, and the highest at 15 °C with 19 d. The highest age-specific fecundities were 2.51 (40 d), 5.42 (22 d), 7.06 (13 d), 6.73 (11 d) and 1.4 (11 d) offsprings at 10 °C, 15 °C, 20 °C, 25 °C and 30 °C, respectively. The peaks of *m_x_* and *l_x_m_x_* tended to increase with the temperature increasing from 10 °C to 25 °C.

The life expectancies (*e_xj_*) of different ages and stages of *A. pisum* at five different temperatures are plotted in Figure 4. Each plot represents the expected survival of an individual at age *x* and stage *j*. The curves showed a decrease with age, although a slight fluctuation was observed at 20 °C. The life expectancy of a newly born nymph was 49.6 d, 36.5 d, 20.7 d, 13.3 d and 6.1 d at 10 °C, 15 °C, 20 °C, 25 °C and 30 °C, respectively.

The age-stage-specific reproductive value (*v_xj_*) shows the contribution of an individual aphid of age *x* and stage *j* to the future population (Figure 5). These curves significantly increased with emerging adults, and the highest peak was observed when females began to produce offsprings at different temperatures. The major peaks in the reproductive values of the females at 10 °C, 15 °C, 20 °C, 25 °C and 30 °C were observed at 27 d (17.54), 13 d (22.78), 10 d (23.70), 8 d (22.11) and 6 d (5.31), respectively. The aphids at 27, 13, 10, 8 and 6 d of age provided the highest contributions to the next generation when reared at 10 °C, 15 °C, 20 °C, 25 °C and 30 °C, respectively.

The derived population parameters are presented in Table 2. The highest net reproductive rate (*R_o_*; 70.9 offspring) was observed at 15 °C, while the lowest *R_o_* was observed at 30 °C (0.67 offsprings). The highest intrinsic rate of increase (*r* = 0.30 d^−1^) and finite rate of increase (*λ* = 1.35 d^−1^) for *A. pisum* were examined at 25 °C. When the temperature increased to 30 °C, both the intrinsic and finite rates of increase dropped, to −0.03 and 0.97 d^−1^, respectively. The longest mean generation time (*T* = 36.3 d) was recorded at 10 °C, which was more than three times the *T* value (11.8 d) at 30 °C.

The population growth and stage structure at different temperatures based on the age-stage, two-sex life table theory are shown in Figure 6. The populations that grew at 20 °C and 25 °C increased significantly faster than those that grew at 10 °C and 15 °C. At 30 °C, the population sizes of the different life stages gradually decreased, which was interpreted as the population becoming extinct in the future. The curves of the stage-specific growth rates of the different temperature treatments approached the intrinsic rate of increase of each temperature condition (0.10, 0.23, 0.28, 0.30 and −0.03 for 10 °C, 15 °C, 20 °C, 25 °C and 30 °C, respectively; Table 2 and Figure 7).

## 4. Discussion

Temperature is one of the most important abiotic factors that affects the development, survival, reproduction and seasonal occurrence of insect populations, because insects are poikilothermic organisms [12,35,36]. We observed the development, survival and reproduction of *A. pisum* in a wide range of temperatures, and analyzed the raw life history data using an age-stage, two-sex life table analysis, showing the population growth and stage-specific growth rates. Our results demonstrated that the life table analysis of *A. pisum* was able to describe the effect of temperature on the population parameters and fitness. The duration of the pre-adult stages significantly decreased with increasing temperature (Table 1). Similar results have been reported for *Rhopalosiphum padi* [35,37,38] and *Tuberolachnus salignus* [39]. The lengths of the immature stages of other aphid species, such as *Nasonovia ribisnigri* [40], *Aphis punicae* [41], *Schizaphis graminum* [42], *Aphis fabae* [26], *Aphis craccivora* [43] and *Brevicoryne brassicae* [44], decreased up to ~25 °C and increased above 25 °C. Total longevity also declined as temperature increased from 10 °C to 30°C, as has also been observed for *A. fabae*, *R. padi*, *A. craccivora* and *B. brassicae*. The age-specific survival rate (*l_x_*) curves sharply dropped at earlier ages when the temperatures ranged from 20 °C to 30 °C. Several studies have also shown the detrimental effects of higher temperatures on the survivorship of various aphid species, such as *R. padi, Toxoptera aurantii, A. fabae, A. craccivora, Sitobion avenae* and *Metopolophium dirthodum* [45,46]. 

The net reproductive rate (*R_o_*) was different for all tested temperatures, and was highest and lowest at 15 °C (70.9) and 30 °C (0.67), respectively (Table 2). The intrinsic rate of increase (*r*) is crucial for determining the growth potential of insect populations under different environmental conditions, and is reflected in the effects of the age of first reproduction, the timing of the peak of reproduction, the duration of the reproductive period, and the survival rate in the population growth rate [47]. In this study, the intrinsic rate of increase and finite rate of increase both increased when temperatures increased from 10 °C to 25 °C, but sharply dropped at 30 °C, to a negative value (*r* = −0.03) and a value less than 1.00 (*λ* = 0.97), respectively. This indicates that the size of the *A. pisum* population reared at 30 °C will decrease compared to the size of the present population as time approaches infinity. Morgan et al. [48] showed that the intrinsic rate of increase of *A. pisum* on *P. sativum* increased with temperatures increasing between 11.9 °C and 23.1 °C on cv. Sancho and 11.9 °C and 19.6 °C on cv. Scout, and declined thereafter. On the other hand, the population of *A. pisum* tested by Mastoi et al. [29] showed that the intrinsic rates of increase were positive values (*r* > 0.26), and finite rates of increase were more than 1.00 (*λ* > 1.30) from 27 °C to 39 °C. This population may have a tolerance to high temperature, and may have been adapted well for fitness at high temperature. 

Lu and Kuo [28] reported that the developmental rate of *A. pisum* in Taiwan increased continuously with increasing temperatures. The duration of the pre-adult period of *A. pisum* reared on *P. sativum* was similar to that observed in our results, but adult longevity was shorter and the number of offspring per female was greater than the values in this study. The survival rate of nymphs at 30 °C (70%) was also much higher than that seen in our study (15%), and nymphs reared at 35 °C successfully developed into adults. However, nymphs reared at 32.5 °C in our study failed to develop into adults (unpublished observation). This study shows that adult aphids were able to produce new nymphs at 30 °C, although the number of offspring was lower than that seen under our other temperature conditions. This result is apparently different from what was observed for the non-reproductive periods at 30 °C with the same aphid species in a previous study [28]. Bieri et al. [49] showed that a high mortality of *A. pisum* occurred, and that aphids reaching the adult stage did not reproduce at 30 °C. The differences between the population parameters of the two populations (i.e., Lu and Kuo [28] and those of our study) were influenced by several factors. Firstly, Lu and Kuo [28] computed the life table parameters using the method of Birch [50], and analyzed the differences in parameters among the different temperatures using Tukey HSD tests based on jackknife estimates of variance [51]. We used an age-stage, two-sex life table analysis, and the statistical differences among treatments were analyzed with a paired bootstrap test at the 5% significance level. Secondly, the host plant as a food source was different between the two studies (i.e., *P. sativum* vs. *V. faba*). Although there was no information regarding the nutritional components of the host plants, the development and reproduction of both populations seem to have been influenced by the host plants. Morgan et al. [48] showed that the pea aphid’s performances (i.e., development time and fecundity) were influenced by host plant cultivars, although Bieri et al. [49] found no differences in the development and fecundity of *A. psium* kept on six different cultivars of peas. Third, the origins of both populations, the different geographical adaptations to temperate and subtropical areas, and the different rearing methods used may have affected the life history data [16,52,53]. Although *A. pisum* populations could develop into adult stages above 30 °C, the populations will decrease or go extinct under those conditions because of the negative intrinsic rate of increase or a lack of reproduction. 

When *A. fabae* were reared with *V. fabae* [26], the pre-adult duration, APOP and TPOP were shorter, while the reproductive period was much longer, than what was seen in our results. In addition, the population parameters, including the net reproductive rate, intrinsic rate of increase and finite rate of increase, at 15 °C, 20 °C, 25 °C and 30 °C, were higher than those of our study. If they have to compete for the same habitat, the *A. fabae* population will be superior to the *A. pisum* population because the population growth of *A. fabae* is greater than that of *A. pisum*. 

Computer programs and computation-intensive statistical methods have been developed to analyze the mass data of biological and ecological studies, including life table analysis [30,34,54]. Taken together, the results of this study suggest that temperature affected the life history data and the derived population parameters of *A. pisum*. Understanding the population parameters and predicting the population growth using life tables can be useful in forecasting population dynamics and establishing mass rearing practices of *A. pisum* for biological control purposes.

## 5. Conclusions

The results of this study provided fundamental information regarding the thermal development of all the life stages, the population parameters, and the population growth of *A. pisum*. *Acyrthosiphon pisum* can survive and reproduce on faba bean plants at all temperatures between 10 °C and 30 °C. The population growth rate of *A. pisum* at 25 °C was greater than under other temperature conditions because the intrinsic rate and finite rate of increase were the highest at 25 °C. The population parameters and projected population growth of *A. pisum* using life tables could be useful for predicting population fluctuations in the field and establishing effective management strategies for *A. pisum*. Future studies may be needed to examine the biological performances of *A. pisum* under fluctuating environmental temperatures, and to forecast the population dynamics. 

## Figures and Tables

**Figure 1 insects-11-00481-f001:**
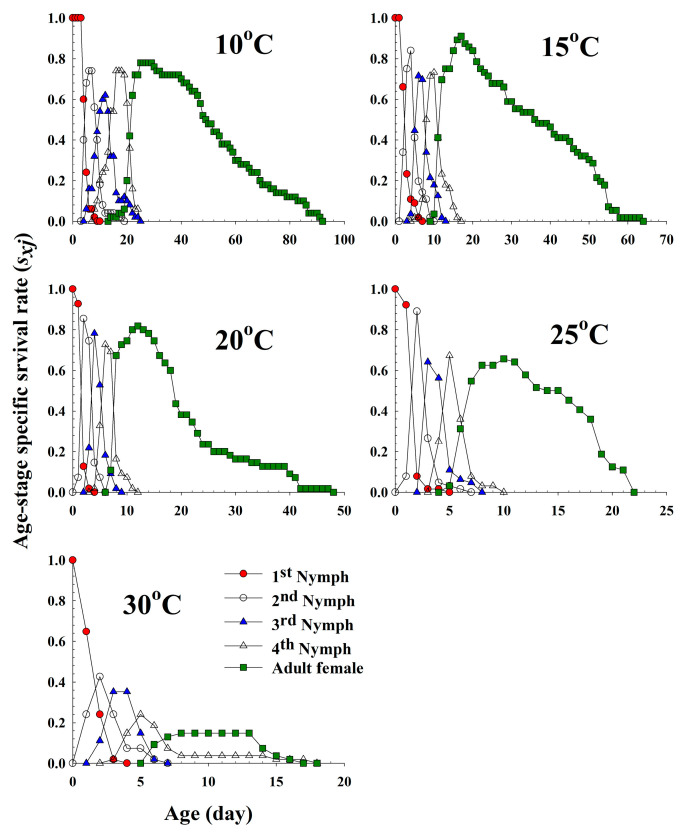
The age-stage-specific survival rate (*s_xj_*) of *Acyrthosiphon pisum* in response to different temperature conditions.

**Figure 2 insects-11-00481-f002:**
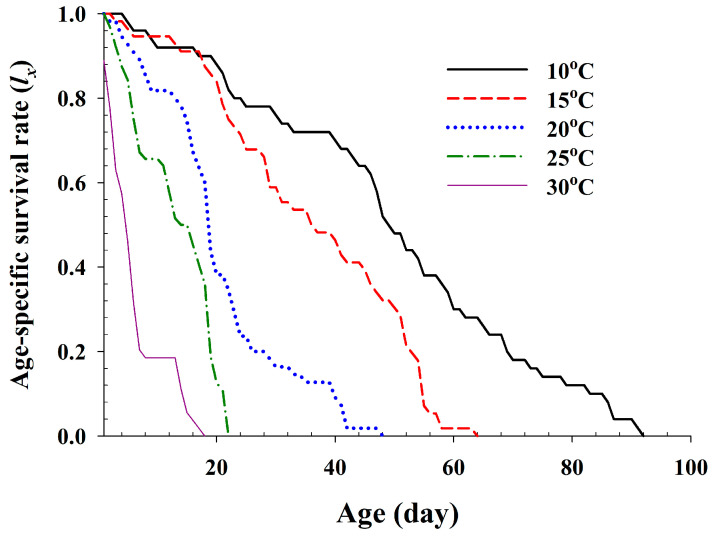
The age-specific survival rate (*l_x_*) of *Acyrthosiphon pisum* in response to different temperature conditions.

**Figure 3 insects-11-00481-f003:**
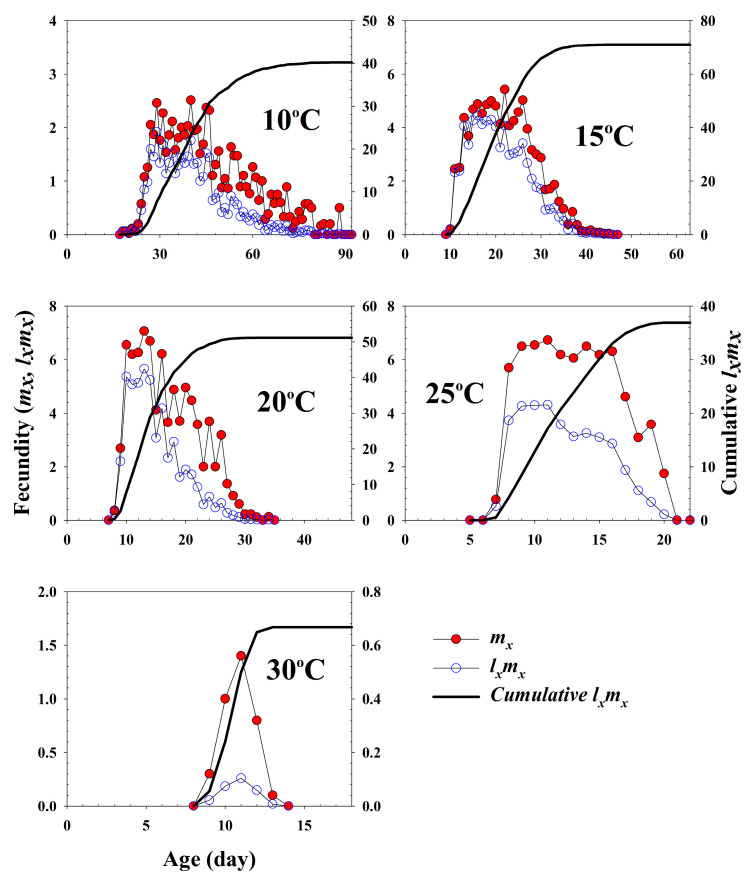
The age-specific fecundity (*m_x_*), the age-specific maternity (*l_x_m_x_*) and the cumulative reproductive rate ($R_{x}=\sum l_{x}m_{x}$) of *Acyrthosiphon pisum* in response to different temperature conditions.

**Figure 4 insects-11-00481-f004:**
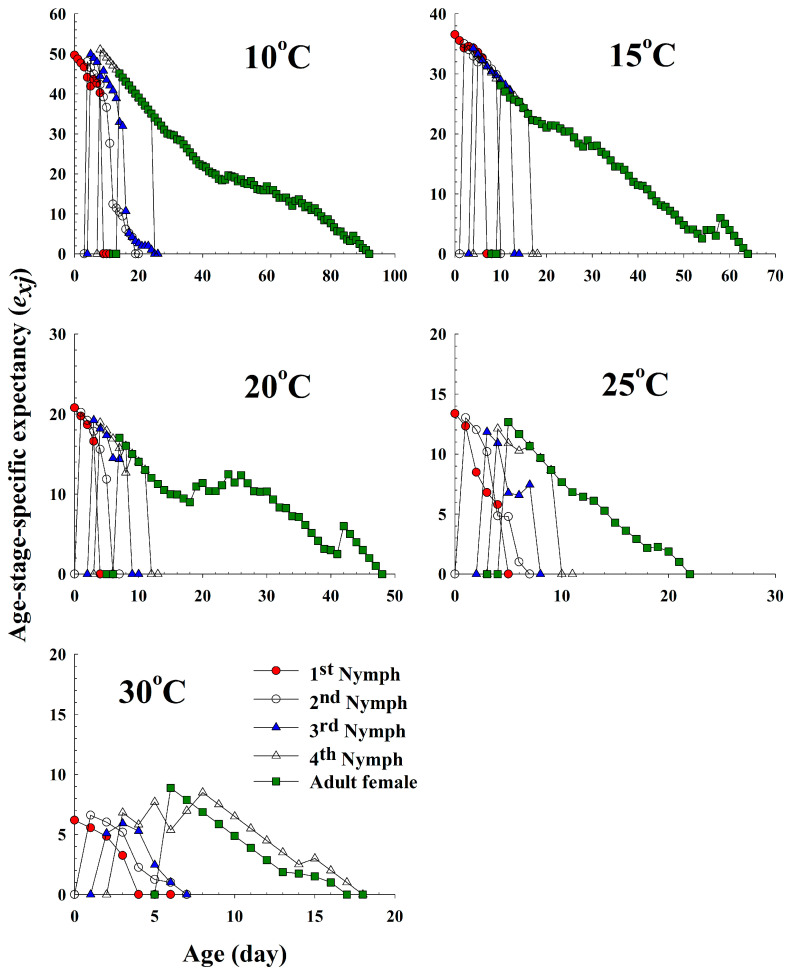
The age-stage life expectancy (*e_xj_*) of *Acyrthosiphon pisum* in response to different temperature conditions.

**Figure 5 insects-11-00481-f005:**
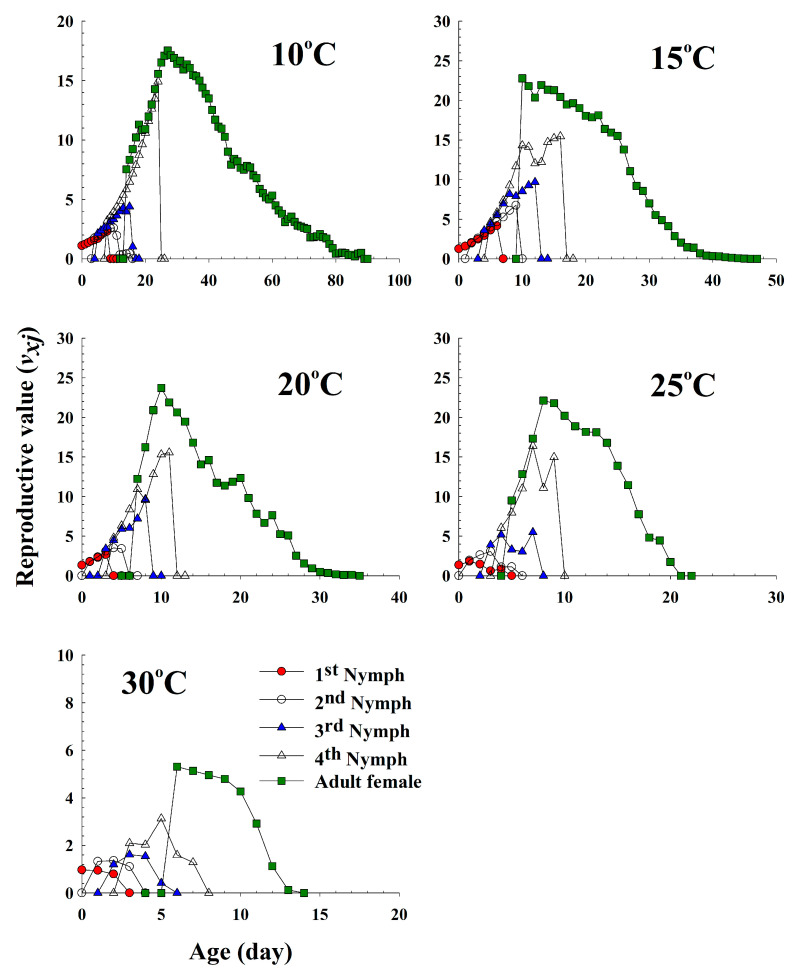
The age-stage-specific reproductive value (*v_xj_*) of *Acyrthosiphon pisum* in response to different temperature conditions.

**Figure 6 insects-11-00481-f006:**
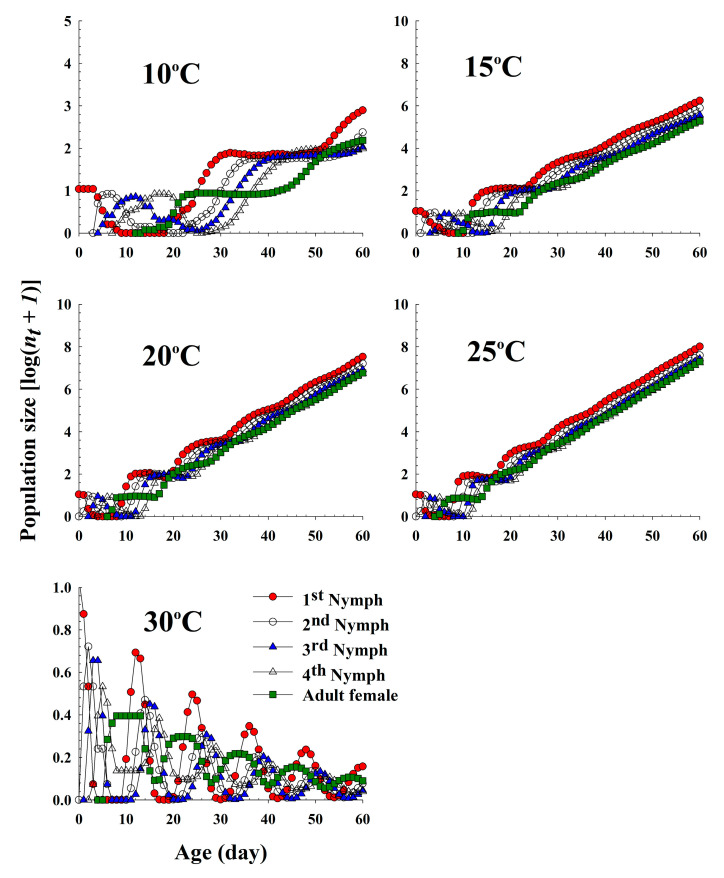
The population growth projection of *Acyrthosiphon pisum* in response to different temperature conditions beginning with an initial population of 10 newly born nymphs.

**Figure 7 insects-11-00481-f007:**
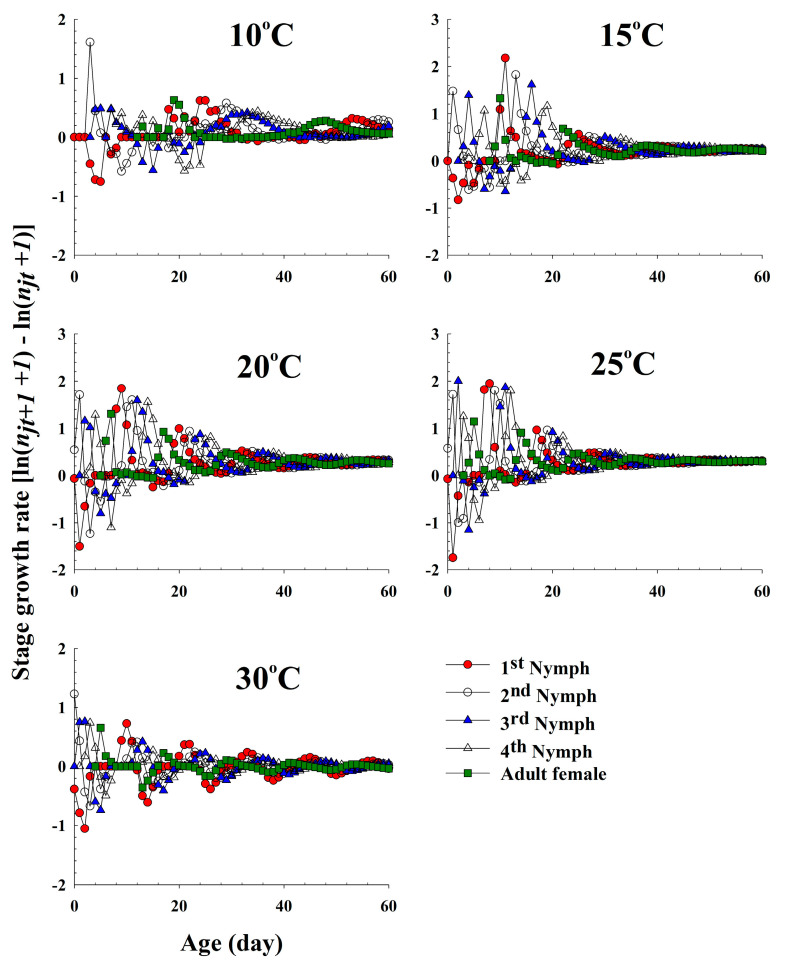
The stage growth rate of *Acyrthosiphon pisum* in response to different temperature conditions.

**Table 1 insects-11-00481-t001:** Developmental time (days), longevity (days), APOP (adult pre-oviposition period, days), TPOP (total pre-oviposition period, days), oviposition duration (days) and fecundity (*Fr*) of *Acyrthosiphon pisum* on faba bean leaves at five different temperatures.

Parameters	Temperature, °C				
	10	15	20	25	30
1st Nymph duration	4.9 ± 0.16 a(48)	3.1 ± 0.16 b(55)	2.1 ± 0.06 c(54)	2.0 ± 0.06 c(61)	1.9 ± 0.13 c(39)
2nd Nymph duration	4.2 ± 0.29 a(47)	2.9 ± 0.12 b(53)	1.9 ± 0.10 c(51)	1.4 ± 0.07 d(56)	1.3 ± 0.08 d(31)
3rd Nymph duration	4.5 ± 0.27 a(39)	2.9 ± 0.11 b(53)	1.9 ± 0.09 c(49)	1.6 ± 0.11 d(49)	1.6 ± 0.12 d(18)
4th Nymph duration	8.3 ± 0.41 a(39)	3.3 ± 0.08 b(53)	2.4 ± 0.09 c(45)	1.9 ± 0.09 d(42)	1.7 ± 0.16 d(8)
Pre-adult duration	21.3 ± 0.31 a(39)	12.2 ± 0.23 b(53)	8.3 ± 0.16 c(45)	6.7 ± 0.16 d(42)	6.5 ± 0.26 d(8)
Adult female longevity	37.7 ± 2.63 a(39)	26.1 ± 1.92 b(53)	15.8 ± 1.34 c(45)	10.9 ± 0.54 d(42)	8.3 ± 0.18 e(8)
APOP	5.1 ± 0.24 a(39)	0.3 ± 0.09 d(53)	1.8 ± 0.14 c(45)	1.4 ± 0.13 c(42)	4.0 ± 0.42 b(8)
TPOP	26.4 ± 0.46 a(39)	12.5 ± 0.24 b(53)	10.0 ± 0.23 c(45)	8.1 ± 0.16 d(42)	10.5 ± 0.42 c(8)
Oviposition duration	22.1 ± 1.65 a(39)	16.4 ± 0.97 b(53)	10.1 ± 0.74 c(45)	8.3 ± 0.53 c(42)	2.0 ± 0.32 d(8)
Fecundity(*Fr*) (nymphs/reproductive female)	51.5 ± 3.93 b(50 *)	74.9 ± 4.24 a(56 *)	62.5 ± 4.22 b(55 *)	56.2 ± 4.11 b(64 *)	4.5 ± 110 c(54 *)
Total longevity	59.1 ± 2.68 a(39)	38.3 ± 1.96 b(53)	24.0 ± 1.35 c(45)	17.7 ± 0.51 d(42)	14.8 ± 0.39 e(8)

Means in the same row followed by different letters are significantly different (*p* < 0.05), as determined by the paired bootstrap test. Numbers in parentheses indicate the number of survivors of *A. pisum* per life stage at each temperature. * in the fecundity row means the number of the initial sample size at each temperature.

**Table 2 insects-11-00481-t002:** Population parameters and proportions of *Acyrthosiphon pisum* reproductive adult females and N-type individuals (those that died in pre-adult stages) at five different temperatures.

Parameters	Temperature, °C				
	10	15	20	25	30
Pre-adult survival rate (%)	0.78 ± 0.05 bc(50)	0.95 ± 0.03 a(56)	0.82 ± 0.05 b(55)	0.66 ± 0.05 c(64)	0.15 ± 0.04 d(54)
First age of survival rate < 50% (days)	49.9 ± 3.17 a(50)	36.3 ± 4.83 b(56)	19.1 ± 0.81 c(55)	14.5 ± 2.01 d(64)	5.0 ± 0.72 e(54)
Net reproductive rate (*R_o_*)	40.2 ± 4.29 bc(50)	70.9 ± 4.60 a(56)	51.2 ± 4.75 b(55)	36.9 ± 4.27 c(64)	0.67 ± 0.26 d(54)
Intrinsic rate of increase (*r_m_*)	0.10 ± 0.003 c(50)	0.23 ± 0.004 b(56)	0.28 ± 0.007 a(55)	0.30 ± 0.01 a(64)	−0.03 ± 0.03 d(54)
Finite rate of increase (λ)	1.11 ± 0.003 c(50)	1.26 ± 0.005 b(56)	1.32 ± 0.009 a(55)	1.35 ± 0.01 a(64)	0.97 ± 0.03 d(54)
Mean generation time (*T*)	36.3 ± 0.69 a(50)	18.4 ± 0.31 b(56)	13.9 ± 0.25 c(55)	12.0 ± 0.17 d(64)	11.8 ± 1.27 d(54)
Proportion of female individuals (*N_f_ /N*)	0.78 ± 0.05 bc(50)	0.95 ± 0.03 a(56)	0.82 ± 0.05 b(55)	0.66 ± 0.05 c(64)	0.15 ± 0.04 d(54)
Proportion of N-type individuals (*N_n_*/*N*)	0.22 ± 0.05 bc(50)	0.05 ± 0.03 d(56)	0.18 ± 0.05 c(55)	0.34 ± 0.05 b(64)	0.85 ± 0.04 a(54)

Means in the same row followed by different letters are significantly different (*p* < 0.05), as determined by the paired bootstrap test. Numbers in parentheses indicate the number of survivors of *A. pisum* per life stage at each temperature.

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
