# Peer review of "Thermal Effects on the Population Parameters and Growth of *Acyrthosiphon pisum* (Harris) (Hemiptera: Aphididae)"

_insects, 2020, doi:10.3390/insects11080481_

Round 1

Reviewer 1 Report

I think this paper is clearly written, with a clean methodology, results, and discussion. The tables and figures are well done and intuitively presented. I have provided comments and suggestions directly in the body of the document using the Adobe Comment tool. I do not have many editorial comments, but I do have some concerns.

First, there is little background of previous studies similar to this given in the introduction. However, the review of previous work is necessary to provide the reader with the 'proof of concept' for any novel study. The authors do a nice job reviewing how previous studies compare with their results in the Discussion, but usually there are similar previous works presented in the introduction. This allows the reader to orient themselves to the field of study and see how the current study differs from others and adds to the body of knowledge of that field. I suggest that the authors add a paragraph in the introduction presenting previous work similar to their study and state how their work differs. I feel this is necessary before the paper can be accepted.

My second concern is this study is very similar to a study recently published in Pakistan Journal of Agriculture Sciences by different authors (Mastoi et al. 2020). Both papers examine effects of five different temperatures on A. pisum reared on V. faba using TWOSEX-MSChart to analyze population parameters (but temperature regimes are different). I have provided the reference below. 

I feel that the similarities in Mastoi et al. 2020 justify it's inclusion in the current paper, in both the introduction and the discussion. In the introduction, Mastoi et al. 2020 should be presented and differences in approach detailed (this relates to my first concern of including a review of similar work in the introduction). In the discussion, a paragraph comparing and contrasting the two papers' results, similar to the paragraph that is currently in it comparing the current study with Lu and Kuo 2008, should be included. Again, this needs to be addressed before the paper is accepted.

To reiterate, the current content of this paper is well written. But, it needs more information on previous similar studies in the introduction and a discussion/comparison of Mastoi et al. 2020 in the discussion.

Reference

Mastoi, A. H., Ahmed, N., Saif-Ur-Rehman, I. A. K., Jiang, J., Hu, X. S., & Liu, T. X. (2020). EFFECT OF HEAT STRESS ON LIFE HISTORY OF PEA APHID, Acyrthosiphon pisum (HARRIS)(HEMIPTERA: APHIDIDAE) BASED ON LIFE-TABLE. Pak. J. Agri. Sci, 57(1), 325-332.

Author Response

All authors really appreciate three anonymous reviewers for valuable comments on our manuscript. Please see the attachment.

Reviewer 2 Report

General comments: this is a well-written study that aims to inform us on the effects of temperature on the life history traits of A. pisum. The topic is very interesting, but I think it needs to be connected to climate change. After all, the experimental design of the different climatic chambers is about different climatic scenarios that may or will occur in nature. Including that in the Introduction part, I think it would better justify the need to have this study and attract more people to read it. Accordingly, the discussion part needs some improvement and some more references so that you can adequately justify the patterns you found. Please see my specific comments below.

-Line 21-22: Please consider deleting “using the age-stage, two-sex life table analysis”

-Line 48: It might worth adding one or two lines here to explain more about this theory.

-Line 66: This is unclear to me. Why did you collect both adults and nymphs? Isn’t the different life stage expected to drive results into a slightly different direction?

-Lines 72-73: It is not clear for how long you preserved the collected individuals in the insectary. It might be an interesting information for some other group that would like to follow the same design.

-Lines 75-77: How many individuals were included in your experiment? I did not see any part where you mention the sample size.

-Line 86: Please replace “ant” with “and”.

-Figure 1: I am a bit confused about the fourth nymph and adult female survival rate as this is shown in the last panel of 30°C. How come the 5rth instar follows the adult female curve? Shouldn’t it stop earlier?

-Lines 256-257: It might worth mentioning in methods that you’ve already tried to rear nymphs at temperatures overpassing the threshold of 32.5°C. This could support better your experimental design.

-Lines 262 – 266: I would not get into so many details about the different statistical approaches. I would rather give some more weight in the next two arguments that follow: that of different food sources and geographical location. You may not have information about nutritional components of host plants but I am sure that you can find some in the literature. In addition, in my experience, the same species but from different localities can show different levels plasticity. It might worth elaborate in a sentence or two and add some references. Have you considered to discuss the “mother” effect? It might be an interesting angle as well.

Sorry, but I don’t see why including “Conclusions”, at least the way it is written now. This part does not give any real conclusion but rather it repeats the most important patterns you found. In my opinion, this should be transferred in the beginning of the discussion and then consider (or not) to rewrite the conclusion section.

Author Response

(The authors gave the same response as above.)

Reviewer 3 Report

In this study, the authors collected life history data of pea aphids at different temperatures and performed life table analysis to determine how the population changes over time at different temperatures. The datasets appear to be of good quality. The analyses seem robust, and the data are consistent with what is seen in other insects. Overall, I found the manuscript to be well written and easy to follow.

I just had a few minor comments:

  • Line 28: In the abstract, you state: “The biological characteristics of A. pisum populations from different geographic areas are also discussed.” I assume this is referring to the study from the population in Taiwan and wonder if this might be misleading. It seems like the rearing and computational methods were different between the two studies, and a direct comparison is not possible. Maybe the authors could consider toning this down a bit and say something like: “Comparisons of our analyses with previous analyses of A. pisum populations from other geographic areas are also discussed”?
  • Line 58: “Although the A. pisum populations evaluated in this study were thelytokous parthenogenesis” sounds weird. Maybe “Although the A. pisum populations evaluated in this study reproduce by thelytokous parthenogenesis”?
  • Line 86: “ant” should be “and”
  • Section 2.4: consider using passive voice
  • Table 1: Define the number in brackets after the means. Also in the sentence, “Means in the same row followed by different letters are significant different”, use “significantly different” instead.
  • Line 208: please add a period at the end of the sentence.
  • Table 2: looks like there are only 5 temperatures even though the title says 6? Also define the number in brackets

Author Response

(The authors gave the same response as above.)

Round 2

Reviewer 1 Report

I reviewed this resubmission and feel it is acceptable for publication. I only had three small changes, and they are embedded in the manuscript.
